# Revisiting the Knowledge Injection Frameworks

**Peng Fu**[*1], **Yiming Zhang**[*1*], **Haobo Wang**[1], **Weikang Qiu**[2], **Junbo Zhao**[1]

[1]Zhejiang University, Hangzhou, China
[2]Yale University, New Haven, America
{p_fu,yimingz,wanghaobo,j.zhao}@zju.edu.cn
weikang.qiu@yale.edu

## Abstract

In recent years, large language models (LLMs), such as GPTs, have attained great impact worldwide. However, how to adapt these LLMs to better suit the vertical domain-specific tasks by utilizing external knowledge remains not completely solved. Indeed, there have emerged a few works on this line where most of them rely on an *alignment* heuristic that is built to inject the corresponding knowledge tuple into the associated text sample.

However, despite the promise, we identify a pivotal problem in this work ubiquitously. Simply put, we find that injecting *unaligned* (i.e., random) knowledge tuple into the LLMs achieves comparable (and sometimes better) results than the aligned knowledge being injected. We therefore take a thorough investigation of this frustrating finding on a variety of related prior work and further provide a chain of potential interpretations for the phenomenon. Based on all that, we offer a simple remediated technique. Briefly, the core of this technique is rooted in an ideological emphasis on the pruning and purification of the external knowledge base to be injected into LLMs. At last, we show that by integrating this technique into most (if not all) knowledge injection frameworks and recent LLMs, it manages to overcome the aforementioned sanity problem and further pushes the boundary of the performance of the domain-adaptive LLMs.

## 1 Introduction

The large language models (LLMs)[1] — like BERT (Devlin et al., 2019), RoBERTa (Liu et al., 2019), GPT-3 (Brown et al., 2020), ChatGPT, GPT-4, etc. — truly have brought gigantic waves worldwide. While these LLMs have evidently extended the frontier of NLP, the prior works (JI et al.,

2022; Bang et al., 2023) have also pointed out that when lacking domain-specific knowledge, LLMs are more prone to hallucinate in downstream tasks.

Notably, a relatively lightweight but promising means to tackle this is through *knowledge injection* such as ERNIE (Zhang et al., 2019), KnowBert (Peters et al., 2019) and K-BERT (Liu et al., 2020) where an external knowledge graph is adopted as shown in Figure 1. Despite the plethora of work on this line being proposed in the past, we present a pivotal problem in this work via comprehensive scrutiny. Generally, this line of work relies on an alignment module where one can automatically associate a given text sample with a knowledge tuple that is extracted from an external knowledge base. This aligned knowledge tuple is then facilitated to influence the downstream task, which manifests a hybrid mixing in the input text (Liu et al., 2020), positional embedding (He et al., 2020) or the upper-level embedding space (Zhang et al., 2019).

**Our findings:** In brief words, for most, (if not all) of the prior work, injecting misaligned, randomized, or (intentionally) irrelevant knowledge tuples yields comparable (and sometimes better) results than the aligned knowledge being injected. More specifically, this ablation protocol indicates a replacement of the matched (aligned) knowledge in Figure 1 by a randomly drafted knowledge. These results are validated both quantitatively and qualitatively on a variety of prominent knowledge injection frameworks across 12 popular datasets. We further note that we dedicate this work to the spectrum of fine-tuning stages thanks to its lightweight nature and arguably wider real-world deployment.

Nevertheless, our work does not mean that knowledge injection is unfeasible as a whole. Rather, the similar mechanism applied in the pre-training stage did have some successes (Ye et al., 2022; Wang et al., 2021b), in spite of the forbidden computational cost incurred. To this end, we believe that there are two prioritized prospects that

---

"*" means both Peng Fu and Yiming Zhang contributed equally to this work.

[1]LLMs are commonly referred to as pre-trained language models (PLMs).

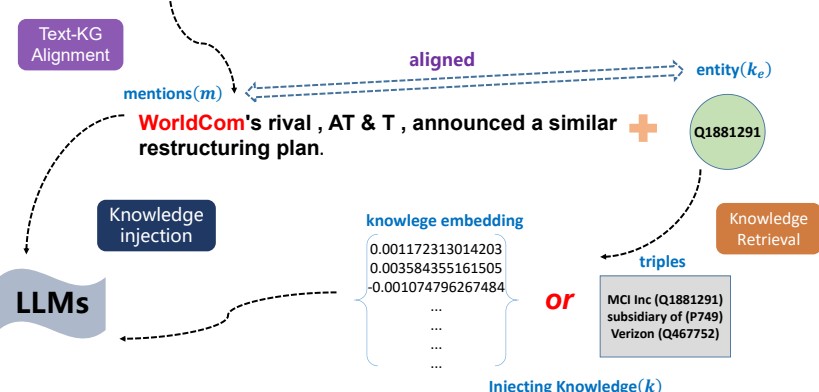

Figure 1: The Process of Common Knowledge Injection. For an input text, the injection method first aligns the entities in the input text (as mentions $m$) with its corresponding entities in the external knowledge base (as entity $k_e$). Afterward, it retrieves the external knowledge that needs to be injected through entity $k_e$. Finally, the injection model injects the input text together with the knowledge into the pre-trained model.

ought to be explained or studied: (i)-to revisit the tuning dynamism[2] by injected knowledge and (ii)-to derive a fix to the problem.

To do that, we first supplement a set of additional experiments by injecting Gaussian noise as a replacement for injected knowledge. Perhaps with surprises, this set of injection flow winds up with indiscernible results with either randomized or aligned knowledge injection. The major question we come up with this far is therefore directed to why the LLMs treat the aligned knowledge similarly to noise. In pursuit of the answer to this question, we cast our hypothesis: Within the fine-tuning scheme, the LLMs fail to adequately disentangle the intricacy possessed in the external knowledge base so as to treat the injected item ubiquitously as noise. For instance, ERNIE (Zhang et al., 2019) intends to integrate a wiki knowledge graph that is composed of a vast of more than 5 million entity nodes. We thereby vaguely connect this hypothesis — together with the prior empirical conclusion — to *data augmentation* that explains why both randomized knowledge and noise injection still renders some performance gain.

At last, rather than composing a complete methodological solution to this newly found problem, we intend to emphasize the importance of injected item itself. In particular, we construct a new conceptual knowledge graph that is purified and pruned from other knowledge base's taxonomy, similar to McCrae et al. (2019). By injecting this

knowledge graph into the aforementioned LLM frameworks, the LLMs work just as expected and manage to overcome the previous sanity-checking experiments. In virtue of this workflow, we posit that our hypothesis is further strengthened and validated. We prove that this remediated technique can seamlessly be consolidated with all prior knowledge injection frameworks, and also recent LLMs such as ChatGPT.

## 2 Related Works

### 2.1 Knowledge injection for LLMs

Recently, the emergence of large pre-trained language models, such as ChatGPT and GPT-4, has attracted great attention from the community and the public, due to their emergent abilities demonstrated in many tasks. Although ChatGPT includes a lot of knowledge through pre-training, the knowledge injection method is still necessary because ChatGPT cannot fully solve problems in professional fields, such as healthcare (Wu et al., 2023; Liu et al., 2023). For this problem, LLMs can pre-train on professional field corpora or retrieve documents (like New Bing) to obtain that knowledge. However, these methods may incur substantial costs and pose a challenge wherein the obtained knowledge may not align seamlessly with the internal knowledge of the models. We aim to integrate the external structured knowledge sources in a more concise and convenient way, rather than updating the internal parameters of LLMs.

---

[2]This refers to the fine-tuning mechanism.

## 2.2 Knowledge-Enhanced Models

Since the large-scale application of pre-trained models in the NLP field, many works expect to improve the downstream tasks' performance by integrating external knowledge. Among those knowledge-enhanced models, many works use knowledge representation-based methods to incorporate factual knowledge (Zhang et al., 2019; Su et al., 2021; Ye et al., 2022; Peters et al., 2019; Wang et al., 2021b; Yamada et al., 2020; Sun et al., 2020; He et al., 2020; Yuan et al., 2021). Other models use other forms to integrate knowledge into the model (Liu et al., 2020; Wang et al., 2021a; Meng et al., 2021; Hosseini et al., 2022,?; Ke et al., 2020; Lu et al., 2022).

Among those works, some achieved eye-catching performances on different downstream tasks. To name a few, **ERNIE** (Zhang et al., 2019) integrates entities' knowledge aligned with the mentions of the input text in the pre-training and fine-tuning stage. **LUKE** (Yamada et al., 2020) proposes an entity-aware self-attention mechanism and forms a multi-way injection summarizing both words and entities. **KnowBert** (Peters et al., 2019) incorporates an additional entity disambiguation module towards improving the entity linker and recombines knowledge features for injection. **K-BERT** (Liu et al., 2020) converts the relation triples with the context into the sentence tree, then encodes them assisted by a novel soft positional encoding method. Although they designed various injection mechanisms, they do not discuss and analyze the research questions in depth. To some extent, this makes their works lack interpretability.

## 2.3 Interpretable Analysis In LLMs

The closest to this work is the transparency and interpretability analysis of knowledge injection frameworks. While there has not been much work covering it, as we go deep into the literature: Peters et al. (2019); Jiang et al. (2020); Cao et al. (2021) have proved that a pre-trained language model can acquire substantial factual knowledge via pre-training on large-scale unlabeled data. Li et al. (2022) analyzes the capacity of LLMs from the aspect of capturing factual knowledge. Zhang et al. (2021) exhibits that injecting redundant and irrelevant knowledge causes an efficiency drop. Hou et al. (2022) shows there is no positive relationship between knowledge injection corpus size and knowledge injection quality.

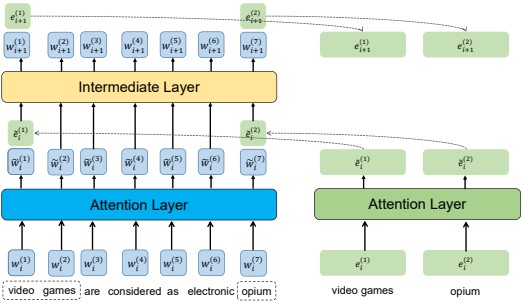

Figure 2: Word And Knowledge Embedding Fusion Process Diagram. Word embedding (blue) and knowledge embedding (green) are usually infused in the intermediate layer, and after that, the related tokens may contain some knowledge-related information, as shown in the iconic framework ERNIE (Zhang et al., 2019).

Regardless, we believe proper study on the transparency of knowledge injection is somewhat critical. This line of work is still at its early stage and is often neglected or deprioritized by prior works. With the study of this work, we may humbly alert the community by showcasing some negative results yielded by our proposed protocol.

## 3 Preliminaries

In this section, we present some preliminary concepts related to knowledge injection. And these introductions serve as a foundation for the subsequent chapters.

### 3.1 Text-KG Alignment

As a prerequisite step, one needs to align the knowledge graph or its subgraph to the input text. For a standard method, it uses the entity alignment tool — such as TagMe (Ferragina and Scaiella, 2010a) — to detect KG's entities mentioned in the input text and link them to the correct KG entry, then tuple them together (Broscheit, 2019). Specifically, given a knowledge graph $\mathbf{G}$ and a sentence $x$, this process can be defined as

$$m, k_e = h(x, \mathbf{G}), \qquad (1)$$

where $m$ denotes entities mentioned in $x$ (mention entity or mentions), $k_e$ denotes the linked entities (a kind of factual knowledge) in $\mathbf{G}$, and $h$ means the entity linking or alignment tool.

### 3.2 Knowledge Injection Methods

From a high-level standpoint, the mission of knowledge injection methods aims to inject an external source of knowledge into the language mod-

| Setup | MedicalNER + MedicalKG F1 | MedicalNER + HowNet F1 | MedicalNER + CnDbpedia F1 | FinancialNER + HowNet F1 | FinancialNER + CnDbpedia F1 |
|---|---|---|---|---|---|
| BERT (w/o KI) | 92.5 | - | - | 86.1 | - |
| K-BERT | 94.2 | 93.3 | 93.8 | 87.3 | 87.4 |
| K-BERT-Aligned | **94.00±0.18** | 93.51±0.15 | 93.48+0.26 | **87.49±0.08** | 87.37±0.19 |
| K-BERT-Random | 93.89±0.33 | **93.52±0.22** | **93.55±0.25** | 87.35±0.19 | **87.46±0.11** |

(a) Results of K-BERT. *K-BERT-Aligned* and *K-BERT-Random* correspond setup 1, 2 respectively. The results BERT and K-BERT come from Liu et al. (2020).

| Setup | BC5chem F1 | BC5dis F1 | NCBI F1 | BC2GM F1 | JNLPBA F1 |
|---|---|---|---|---|---|
| BioBERT (w/o KI) | 92.9 | 84.7 | 89.1 | 83.8 | 79.4 |
| KeBioLM | 93.3 | 86.1 | 89.1 | 85.1 | 82.0 |
| KeBioLM-Aligned | **93.24±0.71** | 87.96±1.05 | 88.46±0.66 | **83.99±0.22** | 78.81±2.51 |
| KeBioLM-Random | 93.06±0.69 | **88.57±0.92** | **88.91±0.25** | 83.25±0.61 | **78.81±2.48** |

(b) Results of KeBioLM. *KeBioLM-Aligned* and *KeBioLM-Random* correspond setup 1, 2 respectively. The results of BioBERT and KeBioLM come from Yuan et al. (2021) and BioBERT is a LLM pre-trained on biomedical corpora.

Table 1: Results of Named Entity Recognition Task. All these experiments are run 5 times with varying random seeds.

| Setup | Open Entity | | | TACRED | | |
|---|---|---|---|---|---|---|
| | P | R | F1 | P | R | F1 |
| BERT (w/o KI) | 76.37 | 70.96 | 73.56 | 67.23 | 64.81 | 66.00 |
| ERNIE | 78.42 | 72.90 | 75.56 | 69.97 | 66.08 | 67.97 |
| ERNIE-Aligned | **78.81±1.05** | 72.15±0.92 | 75.33±0.41 | **71.09±1.62** | **58.15±5.88** | **63.79±3.60** |
| ERNIE-Random | 77.85±1.13 | **73.12±1.07** | **75.37±0.31** | 68.29±5.91 | 58.08±5.83 | 63.73±3.64 |

Table 2: Results of ERNIE on Open Entity and TACRED. *ERNIE-Aligned* and *ERNIE-Random* correspond setup 1, 2 respectively and all these experiments are run 5 times with varying random seeds. The results of BERT and ERNIE come from Zhang et al. (2019). The drop in performance of ERNIE on TACRED may be attributed to the data quality issues inherent in the dataset itself (Alt et al., 2020; Stoica et al., 2021).

els, with an ultimate goal of better suiting the models to downstream tasks, particularly the low-source domains. Throughout the literature, there has emerged a few separate branches shedding light on different paradigms.

To begin with, the major division of this line can be categorized by injection during the pre-training stage or the fine-tuning stage. Hereby, we use the iconic framework, ERNIE (Zhang et al., 2019), for demonstration. On one hand, in the pre-training stage of knowledge injection, ERNIE forms a separate masked language modeling objective. Specifically, it randomly masks off the linked entities and has an additional softmax head to recover it. On the other hand, during the fine-tuning stage, ERNIE fuses the text and aligned knowledge in the vectorial representation space, as shown in Figure 2.

Notice, the purpose of this paper is **not** to propose a novel knowledge injection scheme, nor to promote any existing method. Therefore, abstracting away from one specific showcasing method, we

may use the simplest form to represent the knowledge injection process, as follows:

$$y = f(x, k), \qquad (2)$$

where $x$ denotes the input text, $k$ denotes the injected knowledge regardless of its instantiated form, $y$ denotes the corresponded gold label and $f$ indicates a trainable neural network. Notice, in this investigative work, we cover many instantiations of $f$ and $k$, including not only ERNIE, KnowBert, and other models that integrate external knowledge, but also ChatGPT, GPT-4, etc., mainly for knowledge injection during the fine-tuning stage to achieve optimal empirical transparency.

### 3.3 The Different Injected Knowledge

To explore the above questions, we design a set of ablation experiments with strictly controlled variables. In those ablation experiments, we follow the previous protocol with the origin knowledge-injected models, only changing the knowledge they inject. That knowledge includes:

| Setup | SQuAD 1.1 | |
|---|---|---|
| | Dev Acc | Dev F1 |
| BERT-large (w/o KI) | 84.1 | 90.9 |
| LUKE | 86.1 | 92.3 |
| LUKE-Aligned | **86.22**±**0.37** | 92.34±0.09 |
| LUKE-Random | 86.15±0.15 | **92.39**±**0.11** |

Table 3: Results of LUKE. *LUKE-Aligned* and *LUKE-Random* correspond setup 1, 2 respectively and all these experiments are run 5 times with varying random seeds. The results of BERT-large and LUKE come from Lan et al. (2019) and https://github.com/studio-ousia/luke.

| Setup | WiC |
|---|---|
| | Dev Acc |
| KnowBert-Aligned | **69.53**±**1.24** |
| KnowBert-Random | 69.25±1.09 |

Table 4: Results of KnowBert, *KnowBert-Aligned* and *KnowBert-Random* correspond setup 1, 2 respectively and all these experiments are run 5 times with varying random seeds.

- *Aligned Knowledge:* refers to the retrieved knowledge that is injected into the model, as done by all prior work, described as $k$. The text-Knowledge Graph alignment process is often conducted beforehand.

- *Random Knowledge:* refers to the random selection of a knowledge point from an external knowledge base and using it in the same form as aligned knowledge, denoted as $k_{random}$. No alignment process is conducted.

- *Wiki Triples Knowledge:* refers to the triples extracted from WikiData5M (Wang et al., 2021b), where the knowledge graph only composes entity id from the linked ones $k_e$. The triples are described as $k_1, k_2, \ldots, k_n$, where $n$ represents the length of a triplet matched by an entity ID. Entity-linking is conducted necessarily before retrieving triples.

- *Conceptual Knowledge:* refers to the knowledge extracted from Wikidata and Wordnet, denoted as $k_c$. Specifically, for an entity id in $k_e$, we extract its title and type from Wikidata and find the corresponding concept from Wordnet. Finally, we combine title, type, and concept into a triplet such as (title, type, concept), as the conceptual knowledge of the corresponding entity. The entity-linking process is also necessary to obtain conceptual knowledge.

## 4 Random v.s. Aligned

In this section, we address a research question – *Does the performance improvement of existing injection algorithms truly attribute to the injected knowledge?* To solve this issue, we design a series of ablation experiments with rigorously controlled variables to investigate the practical impact

of knowledge information. The experiment results across 12 pertinent datasets demonstrate that **aligned knowledge injection is *not superior to random knowledge injection.*** In the remainder of this section, we will provide a comprehensive description of the experimental setup and present the corresponding results in detail.

**Ablation Protocol.** In hindsight, the effect of knowledge injection can be decomposed into two parts: (i)-the knowledge injection mechanism as to *how to inject it* and (ii)-the knowledge itself as to *what to inject*. The very majority, if not all, of the prior work is dedicated to the (i) and uses final performance as the sole metric to check if the injection works. Nevertheless, to further enhance the transparency of the system, we wholeheartedly believe that both conditions shall be studied and met. In that regard, to complete the picture, we focus majorly on (ii).

Briefly, we intend to substitute the previously-added knowledge with the random knowledge 3.3, and assess the performance of the original injection (with aligned knowledge 3.3) in comparison. In particular, we adopt the following settings:

1. *knowledge injection* refers to injecting aligned knowledge in the training and testing process, which can be described in Equation 2;

2. *random injection* refers to injecting random knowledge in the training and testing process. Other experimental settings, like baseline and fine-tuning configurations, are consistent with the knowledge injection. It can formally be defined as $y = f(x, k_{random})$;

3. *noise injection* refers to injecting randomized Gaussian white noise in the training and testing process, as $y = f(x, \epsilon)$.

**Backbones and Datasets.** Indeed, different downstream tasks may require different knowledge

| Setup | MedicalNER + MedicalKG | MedicalNER + HowNet | MedicalNER + CnDbpedia | FinancialNER + HowNet | FinancialNER + CnDbpedia |
|---|---|---|---|---|---|
| BERT (w/o KI) | 92.5 | - | - | 86.1 | - |
| K-BERT-Random | **93.89±0.27** | 93.52±0.18 | 93.55±0.20 | **87.35±0.15** | **87.46±0.09** |
| K-BERT-Noise | 93.79±0.32 | **93.73±0.13** | **93.80±0.13** | 87.15±0.11 | 87.10±0.11 |

(a) Results of noise injecting to K-BERT for NER datasets. The results of BERT come from Liu et al. (2020)

| Setup | SQuAD 1.1 Dev Acc | Dev F1 |
|---|---|---|
| LUKE-Random | **86.22±0.37** | **92.34±0.09** |
| LUKE-Noise | 86.09±0.48 | 92.33±0.04 |

(b) Results of noise injecting to LUKE. The results of BERT-large come from Lan et al. (2019)

| Setup | Open Entity P | R | F1 |
|---|---|---|---|
| BERT (w/o KI) | 76.37 | 70.96 | 73.56 |
| ERNIE-Random | **78.81±1.05** | 72.15±0.92 | **75.33±0.41** |
| ERNIE-Noise | 77.28±0.54 | **72.98±0.42** | 75.07±0.06 |

(c) Results of noise injecting to ERNIE. The results of BERT come from Zhang et al. (2019).

Table 5: Results of Gaussian Noise. All these experiments are run 5 times with varying random seeds.

types, scales, or quantities. Distinctive knowledge injection methods and model backbones for different NLP applications may also vary widely. To take a comprehensive revisit of the knowledge-enhanced models, we choose the most advanced as well as the best performing knowledge-injected LLMs as our baselines, in correspondence to the different benchmarks. Following the aforementioned principles, we primarily choose LUKE (Yamada et al., 2020), ERNIE (Zhang et al., 2019), KnowBERT (Peters et al., 2019), K-BERT (Liu et al., 2020) and KeBioLM (Yuan et al., 2021) as the major backbones/methods for the purpose of the study. The methods can be primarily classified into two categories: text-based methods, exemplified by K-BRERT, and embedding-based methods, such as KnowBert, ERNIE, LUKE, and KeBioLM. In the meantime, we cover most of the major datasets. The details and stats of them are provided in Appendix A.1 and A.2. And for the information on knowledge graphs, please refer to Appendix A.3.

**Main Results And Discussion.** Exhibited in Table 1 to 5c, we conclude that: (i) *the knowledge injection is not superior to random injection.* The differences between them generally within 1.0, and some are even lower than 0.1; (ii) the difference between random injection and noise injection is also much neglectable, ranging by no more than 0.3 by F1.

These phenomenons can be further inferred that the knowledge-injected models do not adequately make use of the knowledge injected in the fine-tuning stage, which may be a fatal problem for those injection models. Upon those closer examina-

tions, we have reason to believe that *the model may treat knowledge injection in a way resemblance to white noise injection.*

**Further Analysis.** To further explore the difference between knowledge injection and random injection, we compare their similarity in the encoder and the output of the classifier in Open Entity and TACRED and find the differences are also small. For the details of the further exploration, please refer to Appendix B.

**Takeaways ①.** *Through these experiments, we discover that the previous approaches of knowledge injection, random injection, and even noise injection do not produce notable distinctions. It renders us that they may not be regarded as favorable choices. Drawing from previous analyses, we observe that prior works (Zhang et al., 2019; Yamada et al., 2020) tend to emphasize the injection method itself rather than considering the model's ability to accurately perceive and comprehend the injected knowledge. This could be identified as the underlying cause of the problem.*

## 5 More Does Not Mean Better

Prior results have pointed to a devastating conclusion — the knowledge injection frameworks are not generally grounded in the knowledge injected in the fine-tuning stage. LLMs are more likely to treat knowledge injection in a way resemblance to white noise injection. In this section, to answer the question *why the injected LLMs treat the injected knowledge as noise during the fine-tuning stage*, we begin to analyze the reasons behind this

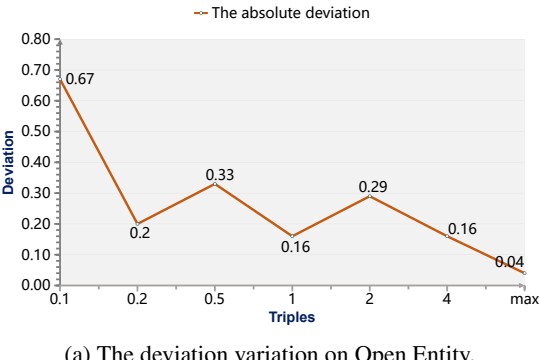

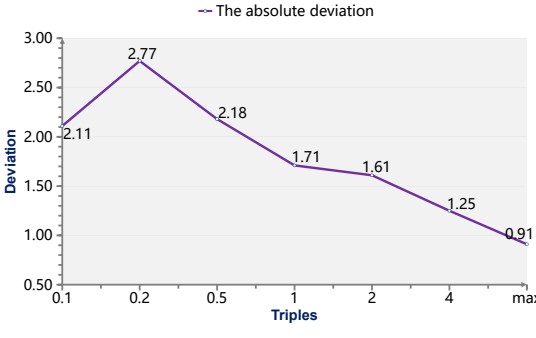

(a) The deviation variation on Open Entity.

(b) The deviation variation on FewRel.

Figure 3: The relationship between the number of injected triplets and the injection effect. The vertical axis represents the F1 deviation between knowledge injection and random injection on the test set. The horizontal axis represents the number of triples extracted from each mention of the input text (for example, if there are 4 mentions in a text, a horizontal axis of 1 means that 4 triples are injected into the text). And 0.1 means that there is a probability of 10% to inject a triplet for each mention.

phenomenon.

As we posit before, excessive and overly complex injecting knowledge may be partially responsible for this issue. To validate our hypothesis, we conducted further experimental explorations by increasing the quantity of relevant knowledge injected at once. An excessive amount of knowledge may not necessarily indicate better performance, and in some cases, it could lead to a performance drop, as per Figure 3.

**Experiment Details.** In previous ablation experiments, there are mainly two types of knowledge injection in the fine-tuning stage, text-based and embedding-based. It is hard to design related experiments in embedding-based knowledge-injected methods, for adding different knowledge embeddings at a single point may lose a lot of information from those knowledge embeddings. However, text-based knowledge-injected methods like K-BERT, which are designed for mass knowledge injection (such as the design of soft-position embedding and seeing layer), are unfit for this kind of experiment.

Based on these, we design a straightforward text-based injection method. This method involves utilizing the corresponding title of the mentions to label the mentioned entity within the text and appending the corresponding triplet of the mentions at the text's conclusion. To give a concrete example, given the original input text, Grumpy Cat, the internet's most famous cat, died at 7 years old. is transferred to be: *Grumpy Cat* Grumpy Cat, the internet's most famous cat, died at 7 years old. (Grumpy Cat type cat). It can be defined as $y = f(x, (k_1, k_2, \ldots, k_n))$, where n is

the amount of injecting knowledge we limited and $(k_1, k_2, \ldots, k_n)$ refer to wiki triples knowledge 3.3. To strictly control the variables and correspond to the previous experimental analysis, we use BERT-base as the baseline and keep the same fine-tuning settings as ERNIE.

**Analysis.** Figure 3 shows the performance difference change between knowledge injection and random injection on Open Entity and FewRel in the text-based method experiment, With the increase of injected knowledge. We could observe that as the number of injected triples increases, the disparity between knowledge injection and random injection diminishes on the whole.

**Takeaways ②.** *More does indeed not mean better without controlling knowledge purity. Consequently, it is imperative to direct our focus from injecting more knowledge to injecting more refined and targeted knowledge.*

## 6 A Remedy by a Simple Method

In this section, we provide an (embarrassingly) simple fix (only in fine-tuning stage) that succeeds in all the aforementioned ablation tests. To alleviate the problem of knowledge injection failure, we introduce conceptual knowledge, which may be more clean and abstract, as a remedy. To validate the effectiveness of the remedy, we devise the last piece of our protocol.

**Injection Details.** In particular, we propose to alter the injected knowledge with a much cleaner and more concise one: $y = f(x, k_c)$, where $k_c$ is the conceptual knowledge 3.3 we construct. In

| Setup | Open Entity | | |
|---|---|---|---|
| | P | R | F1 |
| BERT (w/o KI) | 76.37 | 70.96 | 73.56 |
| ConceptualKI-Random (Ours) | 77.18±0.87 | 73.11±0.61 | 75.09±0.53 |
| ConceptualKI-Aligned (Ours) | **77.53**±1.76 | **73.54**±0.99 | **75.47**±0.44 |

Table 6: Results of Our method on Open Entity. *ConceptualKI-Aligned (Ours)* and *ConceptualKI-Random (Ours)* correspond setup 1, 2 respectively and all these experiments are run 5 times with varying random seeds. The results of BERT come from Zhang et al. (2019).

| Setup | FewRel | | | TACRED | | |
|---|---|---|---|---|---|---|
| | P | R | F1 | P | R | F1 |
| BERT (w/o KI) | 85.05 | 85.11 | 84.89 | 67.23 | **64.81** | 66.00 |
| ConceptualKI-Random (Ours) | 83.54±0.61 | 83.60±0.61 | 83.43±0.62 | **70.87**±0.77 | 54.60±12.14 | 61.04±8.84 |
| ConceptualKI-Aligned (Ours) | **87.47**±0.06 | **87.41**±0.05 | **87.34**±0.06 | 70.81±1.47 | 62.80±2.30 | **66.54**±1.36 |

Table 7: Results of Our method on FewRel and TACRED. *ConceptualKI-Aligned (Ours)* and *ConceptualKI-Random (Ours)* correspond setup 1, 2 respectively and all these experiments are run 5 times with varying random seeds. The results of BERT come from Zhang et al. (2019).

contrast to the factual knowledge base Wikidata (including more than 80 million entities), the conceptual knowledge base Wordnet is significantly smaller, consisting of only 117 thousand concepts. And as the conceptual network (structure of Wordnet) deepens, the conceptual knowledge base can be refined and pruned to a greater extent. With the conceptual knowledge, the example in section 5 is transferred to be: *Grumpy Cat* Grumpy Cat, the internet's most famous cat, died at 7 years old. (Grumpy Cat cat animal).

**Main Results.** We choose BERT-base as our backbone and baseline, and follow the previous protocol. Notice that, among the comparisons, all setups are kept identical except for the different forms of injected knowledge. As shown in Table 6 and 7, we draw the following observations: (i)-the performance difference between correct knowledge injection and random injection has been apparently enlarged compared to previous sections, e.g. +**3.91** F1 on the two relation-extraction datasets; (ii)-this difference on Open Entity remains relatively smaller (**0.32** F1), but it is still better than previous ablation experiment results ($\leq$**0.06** F1). We speculate that this might be caused by the small scale (only 2000 samples for training and testing each) of the dataset.

**Experiment in ChatGPT** To test the practical effectiveness of concept injection in ChatGPT, we extracted some data from TACRED. This experiment was divided into three groups:

1. *Group 1* adopts the text format of paragraph "Injection Details", without triples in the end;
2. *Group 2* retains and injects all the wiki triple knowledge $(k_1, k_2, \ldots, k_n)$, just as $y = f(x, (k_1, k_2, \ldots, k_n))$;
3. *Group 3* injects conceptual knowledge $k_c$, exactly as $y = f(x, k_c)$.

The results demonstrate that both Group 1 and Group 2 exhibit an accuracy level of **88%**. Conversely, Group 3, which incorporates conceptual knowledge, achieves a higher accuracy rate of **92%**, by an absolute **4%** enhancement. It implies that concept injection may exert a discernible impact on ChatGPT. For more experimental information, please refer to Appendix D.1.

**Takeaways ③.** *Pruning the knowledge source is essential for successful knowledge injection into language models.*

## 7 Conclusion

In this article, we present a comprehensive empirical study of current knowledge injection frameworks. Unfortunately, with a series of testing and ablation protocols we propose, most, if not all, prior knowledge injection methods perform erroneously. We then provide an interpretation from the similarity of noise injection. We finally provide a (very) simple remediation method that may remedy the issues. With this work, we wholeheartedly encourage the community towards (i)-further checking the knowledge injection methods; (ii)-focusing a bit more on the side of the knowledge itself, rather than the entire dedication to the knowledge injec-

tion mechanism or the neural architectures. At last, we humbly hope that the set of our protocols can be adapted for sanity-check in future research on this line, together with our simple remediation method applied as an additional baseline.

## Acknowledgement

This work is majorly supported by the NSFC under Grants (No. 62206247), and in part by the National Key Research and Development Program of China (No. 2022YFB3304101). JZ also thanks the sponsorship by the Fundamental Research Funds for the Central Universities (No. 226-2022-00028).

## Limitations

We present our limitations of this work in this section. First, we only pick the most influential, representative and iconic knowledge-injection methods and datasets to form the main body of investigation. Admittedly, there are other works proposed in recent years (introduced in Section 2.2), but that is perhaps beyond the scope of this paper.

Second, we primarily dedicate our extensive study to the knowledge injection performed during fine-tuning. The reasons are three-fold: (i)-knowledge injected within the fine-tuning stage is the most dominant paradigm in a real-world application, compared to the prompting schemes with pre-training which is significantly more unstable; (ii)-knowledge injection in the fine-tuning stage extracts much less computational and carbon cost, so most of the research groups worldwide can freely reproduce our results; (iii)-if we extrapolate into the future of the LLMs, it is trendy that these models' sizes may keep growing. At that point, we believe the portions of the model (say, the first couple layers, some intermediate layers, or the penultimate layers, respectively) can still be fine-tuned and manageable. By contrast, pre-training a whole large LLM with external knowledge-incorporated and/or prompted data would become exponentially harder.

Last but perhaps not least, due to computational limitations, we conduct the relevant experiments only on BERT and RoBERTa models. However, We also provide primary investigation on LLMs, such as ChatGPT. The work's primary objective is to explore and contribute to integrating external knowledge into Language Model architectures. We aim to provide a reference and assistance for future research in knowledge data-centric approaches and inspire future research to render the purity of knowledge.

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

# A  Experiment Details

## A.1  Baseline Details

**LUKE** (Yamada et al., 2020) chooses to enhance RoBERTa with knowledge from Wikipedia. It uses a new pre-training task that involves predicting randomly masked words and entities in a large entity-annotated corpus retrieved from Wikipedia. At the same time, LUKE also inputs wikipedia entities into the model which are based on the sentences in finetuning for the question-answering dataset SQuAD1.1. In addition to injecting knowledge, LUKE proposes an entity-aware self-attention mechanism and considers the types of tokens (words or entities) when computing attention scores (Yamada et al., 2020).

**ERNIE** (Zhang et al., 2019) injects entity knowledge from Wikipedia into BERT in pre-training and finetuning. ERNIE first uses TAGME (Ferragina and Scaiella, 2010b) to link entities mentioned in

context to their corresponding entities in KG, then injects the corresponding entities embedding into language models. Embeddings of the corresponding entities are trained on triples from WikiData via TransE (Zhang et al., 2019).

**KnowBERT** (Peters et al., 2019) integrates knowledge from WordNet and Wikipedia into BERT and demonstrates improved perplexity and ability to recall facts. KnowBERT first trains an integrated entity linker to retrieve relevant entity embeddings, which is used for entity disambiguation. Then, the model uses a Knowledge Attention and Recontextualization (KAR) mechanism to combine the knowledge representation and contextual word representations.

**K-BERT** (Liu et al., 2020) choose CN-DBpedia, HowNet, and MedicalKG as external knowledge bases. K-BERT is devised to feed a structural tree that is decoded from the sentence into a pre-trained language model. The construction of the structural tree is driven by both the sentence itself together with an external knowledge graph. However, it inevitably brings the problem of knowledge noise. To solve this problem, K-BERT proposed to special a seeing layer, which makes the injected triples can only affect their corresponding subject.

**KeBioLM** (Yuan et al., 2021) injects entity knowledge from UMLS (Bodenreider, 2004) by fusing the entities in the knowledge base and mentions in the text in the middle layer. Firstly, it uses a function to recognize if a span is an entity mentioned. then, it links to a set of the mention's k-nearest entities and integrates the entity embedding and the word embedding in the hidden layer, as the input of the model.

## A.2 Downstream tasks and Dataset Details

**Named Entity Recognition (NER)** is the task of finding the corresponding span of the named entity in the given sentence.

Finance NER [3] includes 3000 financial news articles manually labeled, which contain over 65,000 name entities.

Medicine NER [4] is the Clinical Named Entity Recognition(CNER) task that was released in CCKS 2017. The dataset mainly extracts medical-related entity names from electronic medical records.

BC5-chem & BC5-disease (Li et al., 2016) contain 1500 PubMed abstracts that extract chemical and disease entities respectively.

NCBI-disease (Doğan et al., 2014) includes 793 PubMed abstracts that had been detected disease entities.

BC2GM (Smith et al., 2008) is a dataset including 20K PubMed sentences extracting gene entities.

JNLPBA (Collier and Kim, 2004) is a dataset including 2,000 PubMed abstracts that have been identified as molecular biology-related entities.

**Entity Typing** is the task to find the correct type of the corresponding label entities in giving a sentence.

Open Entity (Choi et al., 2018), commonly used in knowledge-enhanced LLMs, has about 6000 sentences with six entity types. Each sentence has five entity labels on average.

**Relation Classification** is the task of identifying the relation between label entities in a given sentence.

TACRED (Zhang et al., 2017), is a relation extraction dataset with 106,264 examples. Examples in TACRED cover 42 relation types.

**Question Answering** is the task of answering questions such as reading comprehension questions.

SQuAD1.1 (Rajpurkar et al., 2016), is a reading comprehension dataset, consisting of questions from Wikipedia articles. SQuAD 1.1 contains 107,785 question-answer pairs on 536 articles.

**Word Sense Disambiguation** is the task to let the model find label words' most suitable entry in the sense inventory.

WiC (Pilehvar and Camacho-Collados, 2019), is a benchmark that is used for evaluating context-sensitive word embeddings. Each instance in WiC has a target word, and the task is to identify if the occurrences of the target word in the two contexts correspond to the same meaning or not.

**Commonsense Causal Reasoning** is the task of finding corresponding options through the causal dependencies.

## A.3 Knowledge graph Details

**CN-DBpedia** (Xu et al., 2017) is a large-scale open-domain encyclopedic Chinese knowledge graph developed by the Knowledge Work Lab of

---

[3] https://embedding.github.io/evaluation/#extrinsic
[4] https://biendata.net/competition/CCKS2017_2/

| Setup | FewRel | | |
|---|---|---|---|
| | P | R | F1 |
| BERT (w/o KI) | 85.05 | 85.11 | 84.89 |
| ERNIE | 88.49 | 88.44 | 88.32 |
| ERNIE-Aligned | 87.98±0.32 | 87.97±0.32 | 87.87±0.33 |
| ERNIE-Random | 85.75±0.26 | 85.73±0.25 | 85.62±0.26 |

Table 8: Results of ERNIE on FewRel. *ERNIE-Aligned* and *ERNIE-Random* correspond setup 1, 2 respectively and all these experiments are run 5 times with varying random seeds. The results of BERT and ERNIE come from Zhang et al. (2019).

Fudan University, covering tens of millions of entities and hundreds of millions of relationships. The CN-DBpedia used in the paper includes 5.17 million triples.

**HowNet** (Dong and Dong) is a large language knowledge base of Chinese vocabulary and concepts, including semantic annotations of Chinese words. The HowNetused in the paper includes 52576 million triples.

**MedicalKG** is the Chinese medical concept knowledge graph, which contains four types of pseudonyms (symptoms, diseases, parts, and treatments). MedicalKG contains a total of 13864 triples and is an open-source part of K-BERT.

**UMLS** (Bodenreider, 2004) is a compendium of many controlled vocabularies in the biomedical sciences. It provides a mapping structure between these vocabularies, containing over 1 million biomedical concepts and 5 million concept names.

**Wiki graph** The knowledge base is Wiki-Data5M (Wang et al., 2021b), which consists of 3085345 entities and 822 relation types.

### A.4 The Results of ERNIE on FewRel

Different from ERNIE's performance on Open Entity and TACRED, the result of ERNIE's ablation experiment on FewRel is about 2.2. However, this result is risky. Because the mention of FewRel is consistent with the task label entity, and the task label is also consistent with the relationship information in Wikidata, the injected knowledge may contain label information (just like the training logic of TransE, the difference between the two entity embeddings is equal to the relationship embedding).

## B Further Explanation

To dive further into these counter-intuitive results, we propose to track down the path of the injected knowledge. As we mentioned in Figure 2, the

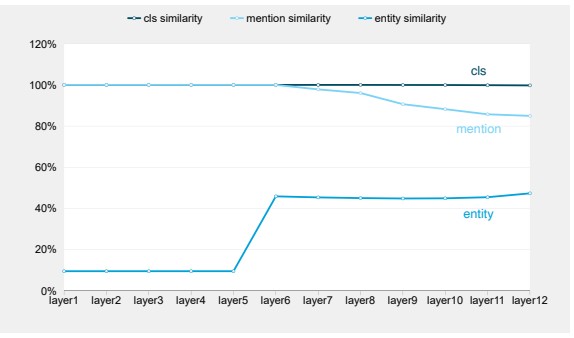

Figure 4: Schematic diagram of embedding similarity change. In this experiment, we inject *aligned* and *random entities* into ERNIE on the Open Entity test dataset. We count the cosine similarity of `[cls]`, mentions and entities embedding in the hidden layers. The similarity in the figure is the absolute average of the 1000 samples.

prior work mostly injects knowledge into hidden layers of the encoder in the representation space. In that regard, we propose to compare the hidden states' similarities. To design this part of the protocol, we prepare a well-trained knowledge-enhanced model, then respectively load the *aligned* and *random knowledge* data feeding through it. In what follows, we calculate the cosine similarity between the two chosen hidden states, specifically, the values of the chosen token's position, from a certain layer, yielded by feeding aligned v.s. random knowledge injection. We mainly focus on the similarities of `[cls]` and mentions' ($m$ defined by Section 1) embeddings, because these embeddings are primarily served as the input gate to our downstream tasks. We choose ERNIE and 2 datasets (Open Entity and TACRED) for this set of experiments.

**Experiment Setup.** We first load the *aligned* and *random entities* data on the trained knowledge-enhanced models and print their output at each hidden layer of the encoder. Then, we compare the similarity among these outputs. We adopt co-

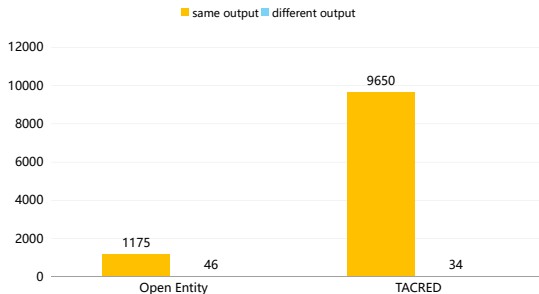

Figure 5: The output stats of injecting aligned and random knowledge. With two inference passes performed, we count the number of samples with the same and different outputs, respectively. Notably, the "same output" (orange bins) indicates that altering knowledge form under our protocol does **not** change the model's prediction, while "different output" (blue bins) indicates the opposite.

sine similarity as a measure of variation in word embeddings and entity embeddings.

Among these similarities, we only keep similarities of `[cls]`, word embeddings related to knowledge, and knowledge embeddings. After this, we output the predictions of the same sentence with different knowledge injected. The result validates the previous inference in fine-grained dimensions.

**Analysis.** Our analytical experiments find that word embeddings injected with different knowledge are highly similar. Figure 4 shows the similarity change of word and entity embeddings from layer 1 to layer 12. From layer 1 to layer 5, there is no interaction between the entity and word embedding, so the similarities did not change. Starting from layer 6, the entity and word embeddings begin to fuse, and the corresponding similarity begins to change, but the `[cls]` embedding changes are always small. After layer 12, `[cls]` embedding inputs into the linear layer and outputs logits.

It can be found that the similarities of `[cls]` embeddings are very high in the whole process, generally above 98%. In this case, it is difficult for the model to find the difference between the three sets of inputs. This result shows that the model hardly obtains valuable information from the knowledge representation.

Since cosine similarity may ignore differences in some dimensions, classifiers may be able to differentiate those differences by eliminating the dominant dimension. So based on the previous experiments, we output the prediction results of the model. Figure 5 shows the results of Open

Entity and TACRED using ERNIE, which loads the test data while injecting *aligned* and *random entities*. Accordingly, it is fairly straightforward to find that it is probably over for the model to have identical predictive results when injected with different forms of knowledge. On TACRED, this portion even exceeds **99.6%**. Simply put, those findings may microscopically explain the reason for the inconspicuous results in previous ablation experiments, that the knowledge-injected models have failed to leverage the injected knowledge, nor to recognize the relevance of the knowledge and the text input.

## C   Performance Gain Explained By Data Augmentation

Indeed, given all the aforementioned empirical results, it is still undeniable that knowledge-injected frameworks have a positive outcome from the perspective of eventual performance. To answer research question 1, we hereby cast a hypothesis, perhaps wild, that the injected knowledge is picked by the model as a data augmentation module.

**Experiment setting.**   To entertain this possibility, we conduct the following two gauges: we meter the degree of overfitting during training that is proximally calculated by the loss gap between the training and validation set. The rest of experimental setup is kept identical to setup 2.

**Discussion.**   In what follows, on the experiments on Open Entity dataset with baseline ERNIE, we curate and report both the loss gap between the train and dev set. From an ordinary machine learning perspective, the larger this gap being revealed, the more overfitted the model gets. The result is summarized in the text as follows: (i)-injecting aligned, unaligned (randomized) or white noise all manage to decrease the loss gap to control overfitting. (ii)-through manipulating the magnitude of the knowledge vector (from 0.173 to 0.170), we can see this gap becomes smaller (but perhaps hurt the overall performance). From an empirical point of view, we may also postulate that these patterns all conform to the data augmentation, such as the regularization effect, the larger scale of augmentation the stronger regularization, etc. At last, as an empirical study, we do not intend to make a deterministic conclusion. The hypothesis we cast — that the previous knowledge injections may act as a imperfect data augmentation module — is based

on their similar performance pattern and perhaps is only one among many other possibilities. We hope to use this finding to motivate the community to provide more theoretical and comprehensive evidence.

## D  Details of Conceptual Knowledge Injection

### D.1  Details of ChatGPT Experiment

In this experiment, we choose 50 data from TACRED, and add the words "Question: Is there a relationship between A and B? If is, what is the relationship between them?" after each text. At last, Group 1 and 2 correctly answer 44 questions, and Group 3 gets 46 correct answers.

## E  Dataset License

We only find three dataset licenses, which are as follows:

SQuAD: CC-BY-SA 4.0
WiC: CC BY-NC 4.0
COPA: BSD 2-Clause