# OpenReview forum: "Revisiting the Knowledge Injection Frameworks"
_EMNLP/2023/Conference — EMNLP 2023 Main_

### Official Review · Reviewer_ic1H · 2023-07-30

**Soundness:** 4

**Excitement:**

4: Strong: This paper deepens the understanding of some phenomenon or lowers the barriers to an existing research direction.

**Paper Topic And Main Contributions:**

This paper investigates whether the previous knowledge injection methods are useful. Through extensive experiments, the author finds that the knowledge injected performs even worse than random knowledge or noise. To deal with the problem, the authors propose a simple remedy method and proven to be helpful.

**Questions For The Authors:**

Does the analysis in the appendix try to say that knowledge injection is hard to work when the injection is done in the hidden layer?
And if so, what's the most effective knowledge injection method? (In text?)
And if not, how to make knowledge injection in the hidden layer work?

**Reasons To Accept:**

1. The investigated problem is important and sheds light on the task of knowledge injection.
2. The writing of this paper is good, making it easy to follow.
3. The experiments are extensive and the conclusion is impressive.

**Reasons To Reject:**

1. There is no clear illustration of the injection method in section 4 (Text-based or Embedding-based?)
2. The remedy method is verified in a text-based injection setting, would it work in embedding-based injection?

**Reproducibility:**

3: Could reproduce the results with some difficulty. The settings of parameters are underspecified or subjectively determined; the training/evaluation data are not widely available.

**Reviewer Confidence:**

3: Pretty sure, but there's a chance I missed something. Although I have a good feel for this area in general, I did not carefully check the paper's details, e.g., the math, experimental design, or novelty.

---

> ### Author Rebuttal · Authors · 2023-08-29
>
> **Q1: There is no clear illustration of the injection method in section 4 (Text-based or Embedding-based?)**
>
> A: We apologize for this oversight.
>
> The experiments provided in Section 4 comprise both Text-based methods, specifically K-BRERT, and Embedding-based methods such as KnowBert, ERNIE, Luke, and KeBioLM, closely following prior related work. Some more information can be found in Appendix A.1.
> We ensure this piece of information will be incorporated in our next revision.
>
> **Q2: The remedy method is verified in a text-based injection setting, would it work in embedding-based injection?**
>
> A: Yes, this is actually a great question, thank you!
>
> We primarily validate our remeditated approach in the text-based injection schemes for its simplicity.
> We supplement a set of expeirments for testing the compatibility of our method with embedding-based injection, as follows.
>
>
> |    FewRel    |     F1     |
> |:------------:|:----------:|
> |     BERT     |    84.89   |
> |  Ours-Random | 85.19±0.24 |
> | Ours-Aligned | 86.03±0.18 |
>
> On FewRel, our remeditated approach makes knowledge injection work properly, by +1.14 over the baseline and +0.84 over the random ablation setup.
> This exhibits similar pattern to the text-based injection shown in Table 7.
>
> We further pose another note tagged with this set of results.
> As we wrote, perhaps the external KG the community has been relying on (all these years) can be the root cause for the problematic performance displayed in the paper.
> Our remeditated approach intends to re-establish it by purifying the KG.
> As per the prior embedding-based injection on this line, this requires us to perform a new round of TransE embedding learning through pre-training.
>
> Here, we believe (i)-there is still much room for improvement and future study here, because the conventional TransE learning may not be optimal for the purified KG;
> (ii)-this new round of training contributes to the final performance comparison as an additional factor, and this is why we left out this verification in the submitted version.
>
> For the time limit, we will complete the same experiments on other datasets in our next revision.
>
>
> **Q3: Does the analysis in the appendix try to say that knowledge injection is hard to work when the injection is done in the hidden layer? And if so, what's the most effective knowledge injection method? (In text?) And if not, how to make knowledge injection in the hidden layer work?**
>
> A: Another great question, thank you.
>
> To the best of our knowledge, there has been few (or none) prior work discussing this issue, about injecting knowledge into the hidden layers.
> The methods in-so-far are mainly categorized into text-based or embedding-based (at the input level).
> On the line of embedding-based injection, we believe that getting the knowledge into the hidden layers properly is far more challenging due to the uncertainty of the positional encoding, higher abstraction-level in the representation, and etc.
>
> From this work, it seems that even injecting knowledge from the input-level embedding is by far problematic, not to mention the more challenging setup (into the hidden layers).
> That said, we believe the investigation of this question is more suitable for a separate future study.
>
> Once again, by doing this work of ours, rather than proposing a new neural network for knowledge injection, we hope to direct the related community to the injected knowledge itself, e.g. a random, unaligned, aligned, or puried knowledge tuple, etc.
> The results from our work encourages a refurbish on current KB/KG adopted in these common benchmarks, and we further provided a remedy to it.

---

### Official Review · Reviewer_jdMq · 2023-08-03

**Soundness:** 4

**Excitement:**

3: Ambivalent: It has merits (e.g., it reports state-of-the-art results, the idea is nice), but there are key weaknesses (e.g., it describes incremental work), and it can significantly benefit from another round of revision. However, I won't object to accepting it if my co-reviewers champion it.

**Paper Topic And Main Contributions:**

In this paper, the authors present a comprehensive empirical study of current knowledge injection frameworks. They find that most, if not all, prior knowledge injection methods perform erroneously. Furthermore, the authors provide an interpretation from the similarity of noise injection. Finally, they offer a simple remediation method to overcome the aforementioned sanity problem and further push the boundary of the performance of the domain-adaptive LLMs.

**Questions For The Authors:**

1. What is the specific implementation of random injection and noise injection?

2. Are there statistics on the number of injected knowledge tuples, including the original and the remediated methods?

3. Can the phenomenon that injecting unaligned knowledge tuples achieves comparable results with the aligned knowledge be attributed to the fact that the aligned knowledge contains too much useless or even noise information?

**Reasons To Accept:**

1. The findings of the authors are meaningful. They find that injecting unaligned (i.e., random) knowledge tuples into the LLMs achieves comparable (and sometimes better) results than the aligned knowledge being injected, which is previously undiscovered.

2. Their findings can encourage the community to further check the knowledge injection methods more from the side of the knowledge itself.

3. The experiments are adequate and detailed, and the experimental results can support the authors’ findings and inferences

**Reasons To Reject:**

1. The contributions of the authors may be limited. They conduct experiments empirically on current knowledge injection frameworks. However, there is not enough research on the underlying reasons behind their findings.

2. Some implementation details are unclear, like the difference in implementing knowledge injection, random injection, and noise injection.

**Reproducibility:**

4: Could mostly reproduce the results, but there may be some variation because of sample variance or minor variations in their interpretation of the protocol or method.

**Reviewer Confidence:**

3: Pretty sure, but there's a chance I missed something. Although I have a good feel for this area in general, I did not carefully check the paper's details, e.g., the math, experimental design, or novelty.

---

> ### Author Rebuttal · Authors · 2023-08-29
>
> **Q1:The contributions of the authors may be limited. They conduct experiments empirically on current knowledge injection frameworks. However, there is not enough research on the underlying reasons behind their findings.**
>
> A: We are sorry that the reviewer may feel this way.
> We’d like to humbly defend our paper.
>
> First, we may argue, devling further into this question would likely to require an establishment of the theoretical scheme of knowledge injection, which is by far unseen from any of preprinted nor published papers.
>
> In current spectrum of the research around knowledge injection, we believe our paper has by far provided one of the most in-depth analysis and dense empirical veirifcation in the community.
> Compared to most of the validation protocol from prior published work, we addtionally offer (i)-a new ablation verification scheme;
> (ii)-a remeditated approach with a new and purifed external KG data/benchmark and
> (iii)-a (potentially) plausible interpretation of drawing similarity to data augmetation, in Appendix C (we are sorry we leave this section to the appendix, we kindly refer the reviewer to it for further proof-read).
>
> Given this context, together with the scope of the paper, we believe it is more suitable for future study.
>
>
>
> **Q2: What is the specific implementation of random injection and noise injection?**
>
> A: Sorry for the confusion.
> Figure 1 in the paper outlines the knowledge injection process that it consists of three steps: text knowledge alignment, knowledge retrieval, and knowledge injection.
>
> During the text knowledge alignment phase, random injection involves substituting the aligned knowledge with random information (usually process during data collation or preprocessing prior to data loading).
>
> In the knowledge retrieval phase, noise injection entails replacing the acquired knowledge embeddings with random white noises (typically perform after input tokenization and before feeding it into the model).
>
> **Q3: Are there statistics on the number of injected knowledge tuples, including the original and the remediated methods?**
>
> A: Appendix A.3 provides the statistical information for several injected knowledge graphs.
> The information as fllowing:
>
> |        KG        |  Entities   | Relations |    Triples    |
> | :--------------: | :---------: | :-------: | :-----------: |
> |    CN-DBpedia    |      -      |     -     | 5.17 million  |
> |      HowNet      |      -      |     -     | 52576 million |
> |       UMLS       |  6 million  |     -     |       -       |
> |    Wiki graph    |  3 millon   |    822    |       -       |
> | Wikidata (ERNIE) | 500 million |     -     |       -       |
> | Wikidata (LUKE)  | 50 million  |     -     |       -       |
> | Wordnet (remedy) |   117,000   |     -     |       -       |
>
> **Q4: Can the phenomenon that injecting unaligned knowledge tuples achieves comparable results with the aligned knowledge be attributed to the fact that the aligned knowledge contains too much useless or even noise information?**
>
> A: Yes, as we posited in the previous response to the Q1, we believe this conjecture very much aligns with ours.
> Again, we may kindly refer the reviewer to the Appendix C.

---

### Official Review · Reviewer_v97Y · 2023-08-09

**Soundness:** 4

**Excitement:**

3: Ambivalent: It has merits (e.g., it reports state-of-the-art results, the idea is nice), but there are key weaknesses (e.g., it describes incremental work), and it can significantly benefit from another round of revision. However, I won't object to accepting it if my co-reviewers champion it.

**Paper Topic And Main Contributions:**

In this work, the authors delve deeply into the adaptation of large language models (LLMs) like GPTs with external knowledge. A thorough empirical study reveals that, under rigorous testing, most contemporary knowledge injection frameworks perform erroneously. This issue is likened to the effects of noise injection. Surprisingly, it's observed that unaligned knowledge can sometimes outperform its aligned counterpart. The paper suggests a straightforward remedy, which emphasizes purifying external knowledge before integrating it.

**Questions For The Authors:**

1. This paper seems to center on BERT-based models. I'm curious about how the findings apply to decoder-only models, such as GPT-2 or Llama?

**Reasons To Accept:**

1. The paper addresses an intriguing topic: the exploration of knowledge injection in language models.
2. The experimental design is robust, providing substantial support for the paper's conclusions.

**Reasons To Reject:**

1. The paper is hard to follow, particularly the introduction. The use of numerous terms, such as "tuning dynamism" makes comprehension difficult.
2. Referring to models like BERT and RoBERTa as "large language models" may be misleading. Typically, LLMs denote language models with over a billion parameters, implying that perhaps BERT and RoBERTa might not be apt descriptors any longer.
3. Attention to detail is needed. For instance, Line 76 has a typo: "successes(Ye et al.,)" – there should be a space after the "successes". Such issues are prevalent throughout the paper.

**Reproducibility:**

4: Could mostly reproduce the results, but there may be some variation because of sample variance or minor variations in their interpretation of the protocol or method.

**Reviewer Confidence:**

3: Pretty sure, but there's a chance I missed something. Although I have a good feel for this area in general, I did not carefully check the paper's details, e.g., the math, experimental design, or novelty.

---

> ### Author Rebuttal · Authors · 2023-08-29
>
> **Q1: The paper is hard to follow, particularly the introduction. The use of numerous terms, such as "tuning dynamism" makes comprehension difficult. Attention to detail is needed. For instance, Line 76 has a typo: "successes(Ye et al.,)" – there should be a space after the "successes". Such issues are prevalent throughout the paper.**
>
> A: We apologize for all of those oversights.
> We promise we’ll fix them in our revision as soon as possible.
>
> **Q2: Referring to models like BERT and RoBERTa as "large language models" may be misleading. Typically, LLMs denote language models with over a billion parameters, implying that perhaps BERT and RoBERTa might not be apt descriptors any longer.**
>
> A: We thank the reviewer for pointing this out.
> In recent days particular, words like Pre-trained Language Models (PLMs), Masked Language Models (MLM) and Large Language Models (LLMs) have been the buzzwords everywhere.
> While we’d like to side with the reviewer on this matter, we did actually scrutinize the past published top-tier papers for these word-usages.
> However, we realized there didn’t seem to have a consensus on naming these models.
> We again are sorry for this confusion, and we will add a note in our revisions.
>
> **Q3: This paper seems to center on BERT-based models. I'm curious about how the findings apply to decoder-only models, such as GPT-2 or Llama?**
>
> A: Yes, this is a great question.
>
> Our response is three-fold:
> (i)-Indeed, we did conducted experiments on ChatGPT using the GPT-3.5-turbo API.
> You may have missed it because we didn’t make a table (sorry!).
> Please checkout Line 494 – 512. The conclusion taken away from this experimetn conforms to the others.
> (ii)-As the reviewer said, we primarily picked BERT, or ROBERTA, as our major backbones.
> This is because most --- if not all --- of the knowledge injection papers [e.g. 1,2,3] live upon them.
> We basically follow the same protocol and setup --- as a commonground --- to make the comparison fair and grouded.
> (iii)-Yes, to have the similar experiemnts launched on GPT-2 or LlaMA would be nice.
> However, here we’d like to ask for grace from the reviewer. As an academical group with limited computing resources, launching these models would be (much) costly.
> There also hasn’t been any highlighted published work studing knowledge injection into these models.
> Thus, on this matter we lacked any contexturelized information including hyperparameter, architecutres and etc.
> To make a full-cycled experiments (explorations of KI on these new backbones, self-run baselines, ablations, our remeditated methods) on these models were unfortunately beyond our resource limit.
>
> *References*:
> [1]Yamada et al. LUKE: Deep Contextualized Entity Representations with Entity-aware Self-attention. 2020
> [2]Zhengyan Zhang et al. ERNIE: Enhanced Language Representation with Informative Entities. 2019
> [3]Matthew E et al. Knowledge Enhanced Contextual Word Representations. 2019

---

### Official Review · Reviewer_Lzwd · 2023-08-12

**Soundness:** 4

**Excitement:**

4: Strong: This paper deepens the understanding of some phenomenon or lowers the barriers to an existing research direction.

**Paper Topic And Main Contributions:**

The paper suggests that current knowledge graph injection approaches are indistinguishable from white-noise injection approaches and propose an "embarassingly" simple solution: refining the knowledge to inject before injecting it. The paper suggests that less is more in this context, meaning that the injection of small amounts of knowledge leads to a treatment different from white-noise-like treatment and a generally better performance of the algorithms.

**Questions For The Authors:**

1) How many epochs did you train for?
2) How did you carry out your hyperparameter optimization?
3) Does the number of dimensions of the embddings impact the performance increase/decrease?
4) Do you results suggest that the current addition of "noise" is basically a regularizer that improves the generalization abilities of the models?

**Reasons To Accept:**

- Very relevant to the community
- Interesting results
- Simple remedy

**Reasons To Reject:**

- Small number of reference language models
- Results might be due to other factors, e.g., the learning parameters

**Reproducibility:**

4: Could mostly reproduce the results, but there may be some variation because of sample variance or minor variations in their interpretation of the protocol or method.

**Reviewer Confidence:**

4: Quite sure. I tried to check the important points carefully. It's unlikely, though conceivable, that I missed something that should affect my ratings.

---

> ### Author Rebuttal · Authors · 2023-08-29
>
> **Q1: How many epochs did you train for?**
>
> A: We are sorry for missing this in our paper.
>
> The train epochs are shown as follows:
> |       dataset       | train epochs |
> | :-----------------: | :----------: |
> | Open Entity, FewRel |      10      |
> |        SQuAD        |      2       |
> |       Others        |      5       |
>
> Presumably, we follow the prior related work’s setting for making the empirical verification fair.
>
> **Q2: How did you carry out your hyperparameter optimization?**
>
> A: Well, the answer is *no*.
>
> We basically follow the prior work’s setting (including the hyperparatmers, architectures, etc.) .
> We didn’t do any extra tuning work on any of the dataset we presented in the experiment section.
>
> **Q3: Does the number of dimensions of the embddings impact the performance increase/decrease?**
>
> A: This is an interesting question.
>
> We actually didn’t know about the answer to this.
> This is because --- as we humbly wrote previously --- that we only adopt the same hyperparamter/architectural-configiguration as the compared method.
> So the dimensions of the embeddings are identical to whichever approach we intended to compared with. However, we intend to supplement some more expeirments in our revision due to the interesting insight this question might bring about.
>
> **Q4: Do you results suggest that the current addition of "noise" is basically a regularizer that improves the generalization abilities of the models?**
>
> A: Yes, we agree with the this might be the cause to explain our findings of this paper.
> In the Appendix C, we put some more analysis intending to interpret this.
>
> Noted, this suspicious high coorelation of the training dynamics --- between prior knowledge injection approach and data regularization --- is also recently supported by [1].
> Despite of our preliminary conjecture and analysis, we believe there are further room to dig especially from a theoretical standpoint. However, that may have gone beyond the scope of this paper. We hope to leave it to future work.
>
> *References*:
> [1] Arora et al. Metadata Shaping: A Simple Approach for Knowledge-Enhanced Language Models. 2022

---

### Meta-Review · Area_Chair_bLCP · 2023-09-17

**Recommendation:** 4

**Metareview:**

This paper studies how knowledge is fused in models, performing an empirical study and comparison of the existing approaches. Furthermore, experiments with randomly injected knowledge demonstrate that these models aren't behaving well as expected and that the misaligned information may result in improvements. Highlighting limitations in this approach for modelling that the community can rally around and address.

The reviews for the paper are generally strong and consistent. Weaknesses identified by the reviewers identify that the paper only contains empirical findings rather than analytical findings and regarding missing information. The missing information is adequately added in the rebuttal and response from the authors and I find the empirical contributions to be sufficient.

---

### Decision · Program_Chairs · 2023-10-07

**Decision:**

Accept-Main

**Comment:**

This paper studies how knowledge is fused in models, performing an empirical study and comparison of the existing approaches. Furthermore, experiments with randomly injected knowledge demonstrate that these models aren't behaving well as expected and that the misaligned information may result in improvements. Highlighting limitations in this approach for modelling that the community can rally around and address.

The reviews for the paper are generally strong and consistent. Weaknesses identified by the reviewers identify that the paper only contains empirical findings rather than analytical findings and regarding missing information. The missing information is adequately added in the rebuttal and response from the authors and I find the empirical contributions to be sufficient.